# The Risk of Major Depressive Disorder Due to Cataracts among the Korean Elderly Population: Results from the Korea National Health and Nutrition Examination Survey (KNHANES) in 2016 and 2018

**DOI:** 10.3390/ijerph20021547

**Published:** 2023-01-14

**Authors:** Min-Jin Kang, Kyung-Yi Do, Nayeon Park, Min-Woo Kang, Kyoung Sook Jeong

**Affiliations:** 1Department of Medicine, Yonsei University Wonju College of Medicine, Wonju 26426, Republic of Korea; 2Department of Preventive Medicine, Yonsei University Wonju College of Medicine, Wonju 26426, Republic of Korea; 3Department of Occupational and Environmental Medicine, Yonsei University Wonju College of Medicine, Wonju 26426, Republic of Korea

**Keywords:** cataracts, depression, Korea National Health and Nutrition Examination Survey (KHNANES)

## Abstract

Cataracts and major depressive disorder (MDD) both have high prevalence, representing for major health burdens globally. In this study, we examined the risk of MDD due to cataracts. Data from the 2016 to 2018 Korea National Health and Nutrition Examination Survey (KNHANES) were used, including 4122 participants. Logistic regression analysis was performed to evaluate the odds ratio for MDD in association with cataracts. Controlled variables were age, gender, smoking, dyslipidemia and mobility. Subgroup analysis was performed with stratification by gender. The results reveal that cataracts are significantly correlated with MDD. Elderly people with cataracts were found to be more likely to develop MDD compared to those without cataracts (adjusted odds ratio: 1.654; 95% CI = 1.197–2.286). In subgroup analysis, men (adjusted odds ratio: 2.631; 95% CI = 1.247–5.551) were found to be more likely to develop MDD than women (adjusted odds ratio: 1.510; 95% CI = 1.061–2.150). Cataracts may be a risk factor for MDD in the elderly, especially among the male population.

## 1. Introduction

Cataracts are an ophthalmologic disease in which the lens become cloudy, causing loss of vision [1]. It is one of the most common causes of visual impairment around the world [1,2]. In 2020, cataracts resulted in roughly 15.2 million cases of blindness in the population over the age of 50 years and 78.8 million cases of moderate-to-severe visual impairment [3]. Various factors are known to contribute to the development of cataracts, including but not limited to aging, smoking, alcohol consumption, high body mass index (BMI), ultraviolet light exposure, diabetes and steroid use [4]. Cataracts can be classified into three groups according to the cause: age-related cataracts, pediatric cataracts and cataracts secondary to other causes, among which age-related cataracts are the most common type in adults, with onset occurring between the ages of 45 years and 50 years [5].

Patients with cataracts often suffer from blurred vision, glare and halos from light. Such ocular symptoms can cause great discomfort and cause stress. A study conducted in 2022 reported that adults with self-reported general vision impairment had higher Kessler psychological distress scores [6]. Moreover, visual impairment due to cataracts may lead to limitations in motility and activities of daily living (ADL), which can negatively affect mental health in the elderly [7]. As such, the association between cataracts and mental health must be closely investigated.

Major depressive disorder (MDD) is one of the most common mental disorders, representing the fourth leading cause of disability [8]. It is characterized by a depressed mood throughout the day and loss of interest in everyday activities. People with MDD may suffer from fatigue, feelings of worthlessness and, in some cases, thoughts of suicide. To date, several studies have identified risk factors for MDD. Depression is more prevalent in in populations that are female [9], unmarried [10], unemployed [11], less mobile [7], consume alcohol [12], smoke [13] and have chronic illness [14]. In particular, the elderly are more likely to suffer from chronic medical conditions, making them more prone to depression. Depression in the elderly is common among those with diabetes [15], stroke [16], myocardial infarction [17] and chronic kidney disease [18]. Although such factors have been determined, ocular diseases have not yet been established as a risk factor for MDD.

Previous studies on the association between cataracts and depression showed conflicting results. A Korean cross-sectional study using the 2008–2012 Korean National Health and Nutrition Examination Survey (KNHANES) database suggested no association between visual impairment and any mental health parameters [19]. On the other hand, a Taiwanese nationwide population-based longitudinal study conducted in 2020 revealed a significant association between cataracts and increased risk of developing depression [20]. In a 2021 Spanish cross-sectional analysis focusing on cataract patients with diabetes, only women were significantly associated with higher odds of depression [21].

Because previous studies were not able to provide definite conclusions, it is necessary to further investigate the relationship between cataracts and depression. In addition, because there are few existing studies comparing male and female populations with respect to the relationship between MDD and cataracts, we would like to examine this aspect as well. The aim of this study is to (1) assess the relationship between MDD and cataracts and (2) further examine the association stratified by gender through a cross-sectional study utilizing the 2016 and 2018 KNHANES.

## 2. Materials and Methods

### 2.1. Participants

This study was based on the 7th KNHANES, which was conducted from 2016 to 2018 using a complex sample design to obtain a sample representative of citizens over one-year-old nationwide. A two-step stratified colony sampling method was applied using the survey district and household as the primary and secondary sampling units, respectively [22]. A total of 16,142 participants took part in this study; 11,700 people under the age of 60 and 320 people without cataracts or MDD examination were excluded. Ultimately, 4122 participants were selected, and the final participation rate was 25.5% (Figure 1). We used open, deidentified data in this study, so exemption was approved by the Institutional Review Board of the Wonju Severance Christian Hospital (CR322343).

### 2.2. Measurements

The KNHANES is an anonymous, self-reported survey that has been conducted annually by the Korea Centers for Disease Control and Prevention since 1998. It targets all Koreans aged 1 year and older residing in the Republic of Korea to determine their current health status and health behavioral trends. The KNHANES database consists of demographic, socioeconomic, nutritional and health-related examinations, including clinical diagnosis and biophysical profiles administered by medical personnel. For specific survey methods, see “Data resource profile: KNHANES” [22]. For the present study, we extracted and used questions from the KNHANES related to cataracts, major depressive disorder, general characteristics and health-related factors.

#### 2.2.1. General Demographic Characteristics

General demographic characteristics comprised age, gender and marital status. Age was categorized into groups of 60–69, 70–79 and 80 years old and older. Marital status was categorized into “married” and “single”.

We examined alcohol and smoking experience, obesity, hypertension, diabetes, dyslipidemia and mobility. Alcohol experience was determined by “yes or no” responses to the question, “Have you ever drunk more than one drink in your life?”. Smoking experience was determined by responding to the question, “How many cigarettes have you smoked in your lifetime?”. Responses of “less than 5 packs” or “more than 5 packs” indicated that the participant has had an experience of smoking in his or her lifetime; a response of “none” indicates that the participant had none. For obesity, physical measurement data from the checkup survey were used. People with a body mass index (BMI) less than 18.5 kg/m^2^ were classified as underweight, those with a BMI of 18.5 kg/m^2^ to 25 kg/m^2^ were classified as normal and those with a BMI of 25 kg/m^2^ and higher were classified as obese.

#### 2.2.2. Comorbidities, Cataracts and MDD

Because hypertension, diabetes, dyslipidemia and mobility are known risk factors of either cataracts or MDD, we included them as confounding factors [15,23,24]. Hypertension was investigated by checking whether a doctor’s diagnosis had been made on a questionnaire. A response of ‘yes’ indicates that the participant has hypertension; a response of ‘no’ indicates that the participant does not have hypertension. The same applies to diabetes, dyslipidemia, cataracts and MDD. Finally, the Euro-QOL 5 Dimension (EQ-5D) is a standardized measure of health-related quality of life and consists of 5 categories: mobility, self-care, usual activities, pain/discomfort and anxiety/depression [25]. For the mobility variable, the category of mobility in EQ-5D was used. Mobility was classified into “no problem in walking”, “some problems in walking” and “confined to bed”.

### 2.3. Statistical Analyses

The study data were analyzed using complex sample analyses. First, we applied the Rao–Scott χ2 test to determine N (weighted %) to examine whether there was a significant difference in the distribution of MDD according to age, gender, marital status, alcohol consumption, smoking, obesity, hypertension, diabetes, dyslipidemia, mobility and cataracts. Next, we performed complex sample logistic regression analysis and determined the odds ratio (OR) and 95% confidence interval (CI) to identify the relationship between cataracts and MDD, with cataracts as the independent variable and MDD as the dependent variable. Model I, which is a crude model, shows the unadjusted OR. Model II shows OR adjusted for age and gender. Model III shows the OR adjusted for age, gender, smoking, dyslipidemia and mobility. Given the known association between cataracts and MDD, we performed subgroup analysis to further examine the association stratified by gender. Statistical analysis was conducted using SPSS (Version: 28.0.1.1). Statistical significance was verified at the *p* < 0.05 level.

## 3. Results

### 3.1. Prevalence of Major Depressive Disorder (MDD) and Cataracts According to General Characteristics and Disease Conditions

Table 1 shows the prevalence of MDD and cataracts according to general characteristics and disease conditions. Among the total of 4122 participants, 3858 people belonged to the non-MDD group and 266 people belonged to the MDD group. The prevalence of MDD was the highest in the 70~79 age group and the lowest in the 60~69 age group (*p* = 0.047). The prevalence of MDD was higher in women than men (*p* < 0.001) and never-smokers were diagnosed with MDD at a significantly higher rate than ever-smokers (*p* < 0.001). Additionally, a higher prevalence of MDD was associated with the presence of dyslipidemia (*p* < 0.001), more severe restrictions in mobility (*p* < 0.001) and cataracts (*p* < 0.001).

Among the total of 4122 participants, 2633 people belonged to the non-cataract group and 1521 people belonged to the cataract group. The prevalence of cataracts increased with age (*p* < 0.001). Significantly more women were diagnosed with cataracts than men (*p* < 0.001), and people who had never experienced drinking alcohol developed cataracts more than those who had drank alcohol (*p* < 0.001). Never-smokers were diagnosed with cataracts significantly more than those who had never smoked (*p* < 0.001). Moreover, a higher prevalence of cataracts was associated with the presence of hypertension (*p* < 0.001), diabetes (*p* < 0.001) and more severe restrictions in mobility (*p* < 0.001).

### 3.2. Odds Ratios of Major Depressive Disorder (MDD) According to General Characteristics and Disease Conditions

Table 2 shows the odds ratios of MDD according to general characteristics and disease conditions. The 70~79 age group was more likely to develop MDD than the 60~69 age group (OR = 1.468, 95% CI = 1.084–1.988). The prevalence of MDD was significantly higher in women (OR = 3.805, 95% CI = 2.568–5.640) and patients with dyslipidemia (OR = 2.109, 95% CI = 1.589–2.800) than in other groups. With respect to mobility, the prevalence of MDD was higher in patients with some problems in walking and patients confined to bed (some problems in walking: OR = 1.887, 95% CI = 1.395–2.55; confined to bed: OR = 5.293, 95% CI = 2.606–10.748) than those with no problem in walking.

### 3.3. Odds Ratios (ORs) of Major Depressive Disorder (MDD) by Cataract

Table 3 shows the risk of MDD according to cataracts and gender. Hypertension and DM were removed as confounding factors, as they showed insignificant results when applied to MDD. In Model I, people with cataracts were more likely to develop MDD (OR = 2.054, 95% CI= 1.551–2.720) compared to those without cataracts. In addition, the differences were significant in Model II (AOR = 1.768, 95% CI = 1.287–2.429) and Model III (AOR = 1.654, 95% CI = 1.197–2.286).

When stratified by gender, both men and women with cataracts were more likely to develop MDD. In Model I, the risk of developing MDD in men (OR = 2.600, 95% CI = 1.245–5.427) was higher than that in women (OR = 1.597, 95% CI = 1.163–2.193). In addition, the differences were significant in Model II (men: AOR = 2.531, 95% CI= 1.221–5.245, women: AOR = 1.624, 95% CI = 1.143–2.307) and Model III (men: AOR= 2.631, 95% CI = 1.247–5.551, women: AOR = 1.510, 95% CI = 1.061–2.150).

## 4. Discussion

We examined the risk of MDD in association with cataracts in South Korean in both gender above the age of 60. Our main findings show that cataracts were associated with increased prevalence of MDD in both men and women. Cataracts were associated with a higher risk of experiencing MDD in both the 70–79 age group and the ≥80 age group than in the 60–69 age group.

According to Korea’s Health Insurance Review and Assessment Service (HIRA) database, cases of cataracts steadily increased from 1,079,836 cases in 2012 to 1,538,520 cases in 2021, and this pattern is likely to continue. The prevalence of MDD in Koreans has also been steadily increasing. The prevalence of MDD increased gradually from 1.6% in 2001 to 2.5% in 2006 and 3.1% in 2011 [26]. Additionally, a study showed an increase in the number of Korean adults with suicidal plans between 2016–2019 and 2020 [27] Both cataracts and MDD are on the rise among Koreans, so more attention is needed to prevent older Koreans with cataracts from developing MDD.

In the present study, elderly Koreans with cataracts were found to have higher odds of developing MDD compared to those without cataracts. One possible explanation for such a result is that a decrease in visual acuity itself can cause emotional distress. In the elderly, those with poor vision show more depressive symptoms than those with normal vision [28]. For this reason, people with visual impairment may be more susceptible to MDD morbidity. Moreover, visual impairment leads to functional loss. Age-related visual impairment was found to be closely related to lower everyday competence [29]. Another study suggested that poorer visual acuity is associated with higher disability in instrumental activities of daily living [30]. As such, functional loss can lead to feelings of helplessness and despair, which could affect mental health and cause MDD. Some studies suggest that cataract surgery has a positive effect on depression in cataract patients [31,32]. When cataracts are removed by surgery, emotional distress and functional loss may disappear, thereby relieving symptoms of MDD.

MDD is commonly known to be more prevalent in women than men; women have 1.7 times higher lifetime prevalence of MDD as a result of various factors. Sociological factors such as poverty, violence and gender inequality contribute to such differences in the prevalence of MDD [33].

Despite the gender differences in the prevalence of MDD, studies focused on the role of gender in the cataract–MDD relationship have been limited. In a Spanish cross-sectional study, female cataract patients showed higher odds for depression, whereas male cataract patients showed insignificant odds [20]. However, such results cannot be applied to general cataract patients, as the database used in the cited study was limited to diabetic patients with cataracts. In our study, the odds of MDD in association with cataracts in men were higher than those in women, indicating that the association between MDD and cataracts is stronger in men than women.

We suggest two main explanations for this result. First, gender differences in reactivity to negative conditions may have contributed to higher odds in men. Under the same negative circumstances, men were found to report more depressive symptoms than equally challenged women [34]. Another study supports this conclusion, suggesting that under specific circumstances, men may be more sensitive to physical discomfort than women [35]. One explanation for this is that men are less likely to find themselves in positions of disadvantage throughout adulthood, so when adversities do occur, men find it more emotionally challenging [36]. The physical discomfort associated with cataracts may lead to significantly higher odds of MDD in men.

Second, visual impairment worsens individuals’ socioeconomic status, increasing the likelihood of being unemployed [37]. Being unemployed could be associated with a loss of identity, leading to feelings of despair and depression. It is more common for men than women to be in a position to earn a living at home by working, so men are more likely to suffer from these effects.

This study has several limitations. First, this is a cross-sectional study, so it is difficult to show the causal relationships between cataracts and MDD. Second, the diagnoses of cataracts and MDD were surveyed based on a questionnaire rather than assessment of objective data such as Korean national health insurance claim data. We did not consider comorbid psychiatric disorders such as anxiety and insomnia. Finally, the questionnaire data were not adjusted for other medical conditions that are highly comorbid with both MDD and cataracts.

Despite these limitations, this study is meaningful, as it is the first to identify the risk of MDD in association with cataracts in both genders.

## 5. Conclusions

We showed that cataracts affect the odds of developing MDD. Therefore, it is important to recognize that cataracts can have effects on mental health in elderly people. As such, it is necessary for family members and physicians to pay closer attention to the mental health of elderly cataract patients, especially men.

## Figures and Tables

**Figure 1 ijerph-20-01547-f001:**
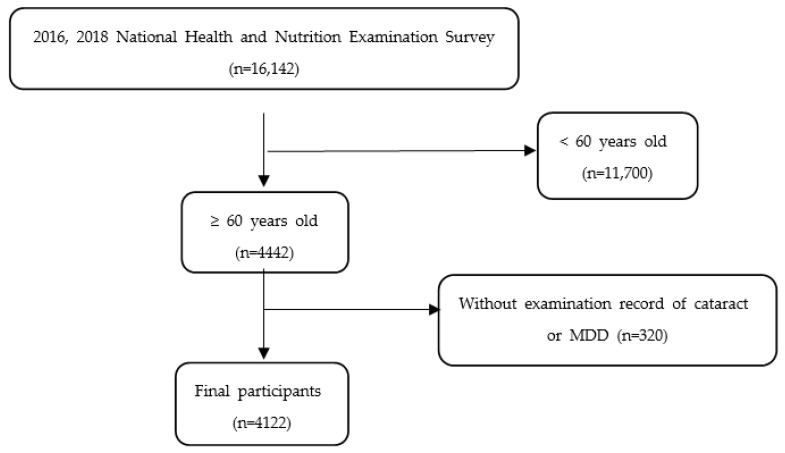
Flow chart of final participants.

**Table 1 ijerph-20-01547-t001:** The prevalence of major depressive disorder (MDD) and cataracts according to general characteristics and disease conditions (N = 4122).

Variable	Total	MDD		Cataract	
		Yes	*p*-Value	Yes	*p*-Value
Total	4122	266 (6.1)		1489 (34.9)	
Age (years), n (%)					
60~69	2047	130 (5.3)	0.047	434 (20.0)	<0.001
70~79	1557	109 (7.6)		752 (49.2)	
≥80	520	27 (5.9)		335 (62.5)	
Gender, n (%)					
Male	1791	46 (2.5)	<0.001	529 (27.2)	<0.001
Female	2333	220 (9.0)		992 (41.8)	
Marital status, n (%)					
Married	4087	263 (6.1)	0.830	1510 (35.3)	0.947
Single	37	3 (7.1)		11 (34.6)	
Alcohol (lifetime), n (%)					
No	894	63 (6.4)	0.093	401 (43.6)	<0.001
Yes	3201	201 (6.0)		1105 (33.1)	
Smoking (lifetime), n (%)					
No	2509	208 (7.7)	<0.001	1029 (40.0)	<0.001
Yes	1579	55 (3.6)		474 (27.7)	
Obesity, n (%)					
Underweight	83	8 (13.9)	0.105	34 (41.7)	0.536
Normal	2394	158 (6.3)		873 (35.2)	
Obese	1553	96 (5.6)		580 (35.3)	
Hypertension, n (%)					
No	2025	117 (5.6)	0.181	599 (28.5)	<0.001
Yes	2099	149 (6.7)		921 (42.1)	
Diabetes, n (%)					
No	3265	210 (6.2)	0.701	1115 (32.8)	<0.001
Yes	859	56 (5.8)		404 (45.3)	
Dyslipidemia, n (%)					
No	2762	127 (4.6)	<0.001	970 (34.3)	0.085
Yes	1362	139 (9.3)		549 (37.5)	
Mobility, n (%)					
No problem in walking	2742	135 (4.8)	<0.001	843 (29.4)	<0.001
Some problems in walking	1303	118 (8.6)		599 (45.8)	
Confined to bed	61	13 (21.0)		40 (65.8)	
Cataract/MDD, n (%)					
No	2633	134 (4.6)	<0.001		
Yes	1489	132 (9.0)			

**Table 2 ijerph-20-01547-t002:** Odds ratios (ORs) of major depressive disorder (MDD) according to general characteristics and disease conditions.

General Characteristics and Disease Conditions	OR	95% CI	
		Lower	Upper
Age (year)			
60~69	Reference		
70~79	1.468	1.084	1.988
≥80	1.125	0.689	1.838
Gender			
Male	Reference		
Female	3.805	2.568	5.640
Alcohol (lifetime)			
No	Reference		
Yes	0.945	0.657	1.360
Smoking (lifetime)			
No	Reference		
Yes	0.441	0.300	0.648
Hypertension			
No	Reference		
Yes	1.211	0.914	1.605
Diabetes			
No	Reference		
Yes	0.931	0.646	1.341
Dyslipidemia			
No	Reference		
Yes	2.109	1.589	2.800
Mobility			
No problem in walking	Reference		
Some problems in walking	1.887	1.395	2.553
Confined to bed	5.293	2.606	10.748

**Table 3 ijerph-20-01547-t003:** Odds Ratios (ORs) of major depressive disorder (MDD) by cataracts.

Group	Model I ^a^	Model II ^b^	Model III ^c^
	Odds Ratio (95% CI)	Adjusted Odds Ratio (95% CI) ^d^	Adjusted Odds Ratio (95% CI)
Total	2.054 (1.551–2.720)	1.768 (1.287–2.429)	1.654 (1.197–2.286)
Male	2.600 (1.245–5.427)	2.531 (1.221–5.245)	2.631 (1.247–5.551)
Female	1.597 (1.163–2.193)	1.624 (1.143–2.307)	1.510 (1.061–2.150)

The data were analyzed by logistic regression for a complex sample. ^a^ Model I: unadjusted OR (95% CI). ^b^ Model II: adjusted for age and gender in total group and age in male and female group. ^c^ Model III: adjusted for covariates in Model II, alcohol use, smoking, hypertension, diabetes and mobility. ^d^ AOR (95% CI): adjusted odds ratio (95% confidence interval).

## Data Availability

Publicly available datasets were analyzed in this study. These data can be found at https://www.kdca.go.kr/yhs/ (accessed on 1 September 2022).

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
