# Peer review of "The Risk of Major Depressive Disorder Due to Cataracts among the Korean Elderly Population: Results from the Korea National Health and Nutrition Examination Survey (KNHANES) in 2016 and 2018"

_ijerph, 2023, doi:10.3390/ijerph20021547_

Round 1
Reviewer 1 Report
This study of 4,000+ individuals from KHANES examines the association between cataracts and depression. The authors find an association (which is not particularly novel), but find meaningful differences by gender (which is novel and important). Major limitations of the article include the selection of covariates (which is highly influential in results here) and description of the definitions of cataracts or MDD. However, this appears to be a meaningful manuscript that would add the existing clinical database of studies and potentially contribute to better understanding risk for depression in older males.
Page 2: Line 80
"Among them, 11,700 people under the age of 60 and 320 people without cataract or MDD 80 examination were excluded"
It is surprising to me that over 90% of participants older than 60 had either cataracts or MDD. What is the breakdown of individuals with MDD, with cataract, and both?
A major drawback to this manuscript is that MDD and cataract are not defined. How was MDD diagnosed? Was this development ever of a cataract or a current cataract? More explanation needed.
I see the models listed and appreciate the examining of different covariates - however, why would education level be associated with cataracts or MDD. I don't think there is any direct link other than via other factors (e.g., access to healthcare, seeking healthcare). This is likely via other means,e.g. poverty or overall health status. This should not be a covariate just because it is significant.
Per review of the literature, age, DM2, smoking, obesity, HTN, and excess alcohol use are variables most strongly associated with cataract development (and also with MDD). However, how does model III change if education level, dyslipidemia, and mobility are removed and replaced with obesity, hypertension, and alcohol use. And how do the results alternatively change when substituting overall total medical combordities (which are associated with MDD/cataracts as well).
I agree with the breakdown by gender, but a better justification needs to be provided. I don't know of any, but the fact that this is not known would be a sufficient justification.
For analysis 3.3, HTN and DM should not be removed from models - these are both associated with cataracts.
More elucidation in discussion is needed of why cataract development is associated with increase in MDD to a greater degree in males than females. Most medical conditions will be associated with MDD - but what is unique to this study (and important clinically) is the above finding.
A major limitation of this research is that the questionnarie data likely could not adjust for other medical conditions that are highly comorbid with both MDD and cataracts. This should be mentioned or at least added as a covariate.
Reviewer 2 Report
Dear the authors,
Can you propose the possible mechanisms that can be the causal relationships between cataract and MDD in discussion?
